# Reaction scope and mechanistic insights of nickel-catalyzed migratory Suzuki–Miyaura cross-coupling

Yuqiang Li[1], Yixin Luo[2], Long Peng[1], Yangyang Li[1], Binzhi Zhao[1], Wang Wang[1], Hailiang Pang[1], Yi Deng[1], Ruopeng Bai[2], Yu Lan[2,3]* & Guoyin Yin [1]*

Cross-coupling reactions have developed into powerful approaches for carbon–carbon bond formation. In this work, a Ni-catalyzed migratory Suzuki–Miyaura cross-coupling featuring high benzylic or allylic selectivity has been developed. With this method, unactivated alkyl electrophiles and aryl or vinyl boronic acids can be efficiently transferred to diarylalkane or allylbenzene derivatives under mild conditions. Importantly, unactivated alkyl chlorides can also be successfully used as the coupling partners. To demonstrate the applicability of this method, we showcase that this strategy can serve as a platform for the synthesis of terminal, partially deuterium-labeled molecules from readily accessible starting materials. Experimental studies suggest that migratory cross-coupling products are generated from Ni(0/II) catalytic cycle. Theoretical calculations indicate that the chain-walking occurs at a neutral nickel complex rather than a cationic one. In addition, the original-site cross-coupling products can be obtained by alternating the ligand, wherein the formation of the products has been rationalized by a radical chain process.

[1] Institute for Advanced Studies, Wuhan University, Wuhan 430072 Hubei, PR China. [2] School of Chemistry and Chemical Engineering, Chongqing Key Laboratory of Theoretical and Computational Chemistry, Chongqing University, Chongqing 400030, PR China. [3] College of Chemistry and Molecular Engineering, Zhengzhou University, Zhengzhou 450001, PR China. *email: lanyu@cqu.edu.cn; yinguoyin@whu.edu.cn

Transition metal-catalyzed cross-coupling reactions have developed into powerful approaches for carbon–carbon bond formation, and have revolutionized synthetic strategies in medicinal chemistry and material science[1–5]. Historically, the successful cross-coupling of alkyl electrophiles represents a landmark of this research field[6–9]. The β-hydrogen-containing, electronically unactivated alkyl partners is generally recognized to be more challenging than their aryl and vinyl analogs[10–12]. Over the past two decades, significant effort has been devoted to avoiding β-hydride elimination and constructing chemical bonds at the original position in an efficient and selective manner[13–16]. Chain-walking highlights great opportunities in bond formation at new positions, the exploration of transformations involving chain-walking have stimulated considerable interest in the synthetic community recently[17,18]. In this context, the concept of migratory cross-coupling has been initially described by the Baudoin group in palladium-catalyzed C–H bond functionalization reactions (Fig. 1a)[19–21].

Recently, it has been reported that nickel catalysis[22] exhibits unique performance in migratory transformations[23]. Particularly, a set of reductive migratory cross-coupling with aryl halides have been disclosed by Zhu[24,25] and our group[26,27] independently, wherein a benzylic selectivity is achieved and a series of pharmaceutically relevant 1,1-diarylalkanes can be efficiently constructed (Fig. 1b). In the past decade, extensive efforts have been devoted to the mechanistic studies of nickel-catalyzed cross-coupling of alkyl electrophiles[13,14,28–30]. While the well-known Ni (0/II)[10,31,32] or Ni(II/IV)[8,33] catalytic cycles are proposed in some cases, a radical chain process and bonding formation from a Ni (III) intermediate is particularly prevailed in those reactions with

a nitrogen-based ligand[29,30,34]. In contrast, there are still quite limited reports on the mechanism of the nickel-catalyzed migratory cross-coupling reactions[25,35]. Compared with classical cross-coupling reactions, the major difference of migratory cross-coupling is a metal chain-walking process involved in the catalytic cycle. More elementary steps in catalytic cycle mean more complicated in mechanism, which increases difficulty into the mechanistic investigations as well. As a result, mechanistic information on the details for catalytic cycle, nickel chain-walking process and the origin of regioselectivity and so on, is still scarce, which seriously restricts further reaction design.

Furthermore, the current methods have several limitations in their substrate scope. First, in our previous investigations, the aryl coupling partners were restricted to electron-neutral and -rich aryl bromides[26,27]. Second, functionalization at benzylic position after chain-walking has been well-studied, but the synthetically useful allylic positions are unachievable[24,26]. Besides, the use of inexpensive, abundant nucleophiles for a redox-neutral migratory coupling reaction remains underdeveloped. For example, the migratory Suzuki–Miyaura cross-coupling of alkyl electrophiles has only been demonstrated by Sigman with palladium catalysis, wherein only allylic products and 1,2-palladium migration could be efficiently achieved (Fig. 1c)[36,37]. Moreover, the especially useful, unactivated alkyl chlorides[38–40] have not been successfully employed in a migratory coupling event to date.

Here, we report our work on developing the general, Ni-catalyzed migratory Suzuki–Miyaura cross-coupling of unactivated alkyl electrophiles with both aryl and vinyl boron reagents (Fig. 1d). Experimental and computational studies are performed to reveal the detailed mechanism.

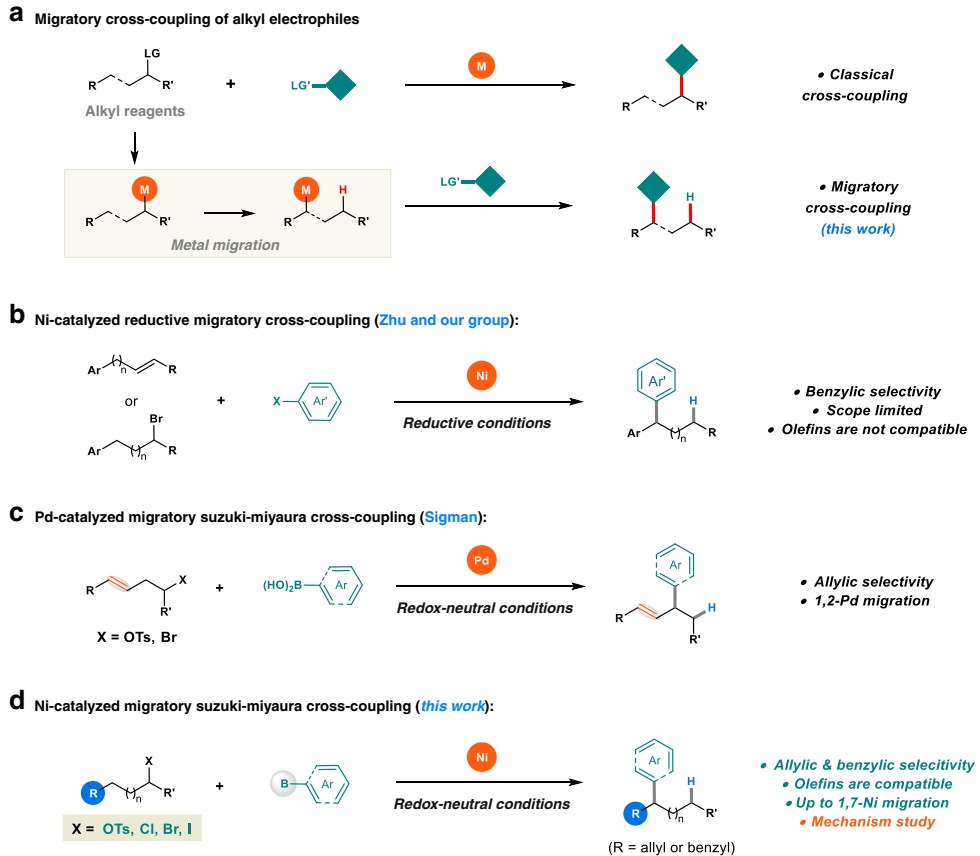

**Fig. 1 Transition metal-catalyzed migratory cross-coupling. a** Migratory cross-coupling of alkyl electrophiles. **b** Ni-catalyzed reductive migratory cross-coupling. **c** Pd-catalyzed migratory Suzuki–Miyaura cross-coupling. **d** The approach developed in this study.

## Table 1 Reaction optimization[a].

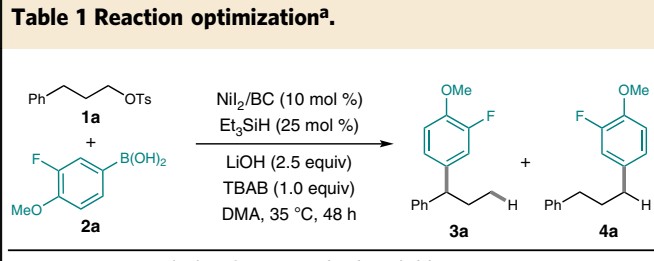

| Entry | Deviation from standard conditions | Yield, % | 3a:4a |
|---|---|---|---|
| 1 | No | 89 (82)[b] | 27:1 |
| 2 | No Et$_3$SiH | 6 | 4:1 |
| 3 | Zn instead of Et$_3$SiH | 35 | 12:1 |
| 4 | ZnMe$_2$, MeMgBr instead of Et$_3$SiH | Trace | – |
| 5 | KF, Na$_2$CO$_3$, Cs$_2$CO$_3$, NaOH instead of LiOH | Trace | – |
| 6 | No TBAB | 8 | 2:1 |
| 7 | NiCl$_2$ instead of NiI$_2$ | Trace | – |
| 8 | NiBr$_2$ instead of NiI$_2$ | 46 | 21:1 |
| 9 | Ni(COD)$_2$, no Et$_3$SiH | 12 | 2:1 |
| 10 | L1 | 28 | 2:1 |
| 11 | L2 | 81 | 51:1 |
| 12 | L3 | 78 | 35:1 |
| 13 | L4 | 76 | 1:77 |
| 14 | NiI$_2$/BC (5 mol%) | 72 | 19:1 |
| 15 | 60 °C, 24 h | 71 | 8:1 |
| 16 | 60 °C, no Et$_3$SiH | 10 | 6:1 |

[a]Standard conditions: NiI$_2$ (15.6 mg, 0.05 mmol, 10 mol%), **BC** (18.0 mg, 0.05 mmol, 10 mol%), Et$_3$SiH (20 μL, 0.13 mmol, 25 mol%), **1a** (145.0 mg, 0.5 mmol, 1.0 equiv), **2a** (127.5 mg, 0.75 mmol, 1.5 equiv), LiOH (29.9 mg, 1.25 mmol, 2.5 equiv), TBAB (161.2 mg, 0.5 mmol, 1.0 equiv), DMA (4 mL). Yields were determined by GC with 1,3,5-trimethoxybenzene as the internal standard
[b]Isolated yield

**BC:** R = Ph, R$^1$ = R$^2$ = Me
**L1:** R = Ph, R$^1$ = Me, R$^2$ = H
**L2:** R = H, R$^1$ = R$^2$ = Me

**L3:** R = H, R$^1$ = R$^2$ = Me
**L4:** R = Me, R$^1$ = R$^2$ = H

## Results

**Reaction optimization**. We launched the investigation with the cross-coupling of 3-phenylpropyl tosylate (**1a**) with aryl boronic acid **2a**. The preliminary study demonstrated that the generation of active nickel catalyst was not efficient via reduction of Ni(II) salts by **2a**. A series of Ni(II) precatalyst activation methods were tested in the following studies[41]. The addition of 25 mol% Et$_3$SiH, proved to be an efficient method to generate the active nickel catalyst, which offering the coupled products in 82% isolated yield (Table 1, entries 1–5, and Supplementary Fig. 3). Base is also crucial to the success of this transformation. The importance of LiOH was highlighted by the failure of other bases which only afford trace amount product (Table 1, entry 5, and Supplementary Table 1).

The addition of TBAB greatly improves the efficiency, which due to the in situ conversion of the alkyl tosylate into alkyl bromide (Table 1, entry 6). Nickel iodide is superior to other nickel precatalysts (Table 1, entries 7–9). Dimethyl substitution on 1,10-phenanthroline or 2,2′-bipyridine backbones are crucial for the migratory selectivity (Table 1, entries 10–13). Notably, the original-site Suzuki–Miyaura cross-coupling[42,43] product **4a** could also be obtained in a good yield and excellent selectivity when the 5,5′-dimethylbipyridine (**L4**) was used (Table 1, entry 13), which is consistent with our previous reductive cross-couplings[26]. Solvent

evaluation indicated that only amide solvents could deliver the migratory product **3a** (Supplementary Table 1). Decreasing the catalyst loading to 5 mol% resulted in moderately lower yield and selectivity (Table 1, entry 14). Increasing the temperature to 60 °C led to worse selectivity (Table 1, entries 15 and 16).

**Substrate scope**. With the optimal conditions in hand, we next shifted our attention to investigating the generality of the Ni-catalyzed migratory Suzuki–Miyaura cross-coupling reaction. First, the scope of alkyl tosylates, which are readily accessible from the corresponding alcohols, was examined. As shown in Fig. 2 (top part), all examples furnished the migratory coupling products in good to excellent yields and regioselectivity under the standard conditions. A class of 1,1-diarylalkanes was prepared accordingly, which are widely encountered in biologically relevant molecules[44,45]. To our delight, both electronic-rich and deficient aryl coupling partners performed smoothly under this redox-neutral conditions. Moreover, homoallyl (**3v**) and homostyrenyl tosylates were also suitable substrates for this nickel-catalyzed system. Vinyl (**3m** and **3×**) and styrenyl (**3w**, **3ae-3ag**) boronic acids coupled with the homostyrenyl tosylates, to provide the skipped diene products with high regioselectivity (**3×**, **3ae-3ag**). It should be noted that in Knochel's study, an olefin coordination with nickel could facilitate reductive elimination at the original position[32].

The scope of the nickel-catalyzed migratory reaction could also be extended to alkyl halides. As illustrated in Fig. 2 (**3ah–3e**), both primary and secondary alkyl bromides were able to yield the benzylic coupling products efficiently under the conditions without TBAB. The tertiary bromide was also able to provide the migratory product, albeit in low yield under the current conditions (**3ap**). The coupling reaction could also proceed when the iodide analog was employed, but with poor regioselectivity under the same conditions (**3as**). Gratifyingly, the more challenging chloride electrophiles were also converted to the benzylic arylation products efficiently in the presence of KI under an elevated temperature (80 °C) (Supplementary Table 2), which dramatically broadens the application of this migratory strategy in cascade or iterative synthesis with polyhalogenated materials[46]. In addition, a gram-scale experiment showed the scalability of this method (**3aj**). All 1,2- to 1,7-Ni migration (**3s**) coupling products could be successfully accessed under the redox-neutral conditions. Finally, a range of functional groups such as trimethylsilyl (SiMe$_3$), ester (CO$_2$Me), amines (NMe$_2$, NPh$_2$), thioether (SMe), cyano (CN), double bonds (C=C), pyridinyl, and quinolyl were well tolerated in this transformation. Notably, the double bonds can survive in this protocol makes it different from the strategy of using olefin and stoichiometric silanes as the susbstrates[25].

Aryl boronic esters could also be employed as the coupling partner to deliver the migratory coupling products (Fig. 3a). It is noteworthy that 3,5-disubstituted aryl boronic esters can be prepared directly from the corresponding arenes by a highly efficient iridium-catalyzed borylation[47].

In addition, the scope of the original-site Suzuki–Miyaura cross-coupling of alkyl bromides with aryl boronic acids was also studied. As shown in Fig. 3b, the electronic property of the aryl boronic acids did not affect the reactivity. Both primary and secondary alkyl bromides could successfully yield the corresponding cross-coupling products. However, tertiary alkyl bromide shows no reactivity. It should be noted that the migratory benzylic cross-coupling products were also detected in most cases, but no terminal coupling products were observed in the reactions with secondary alkyl bromides due to the limitation of our instruments.

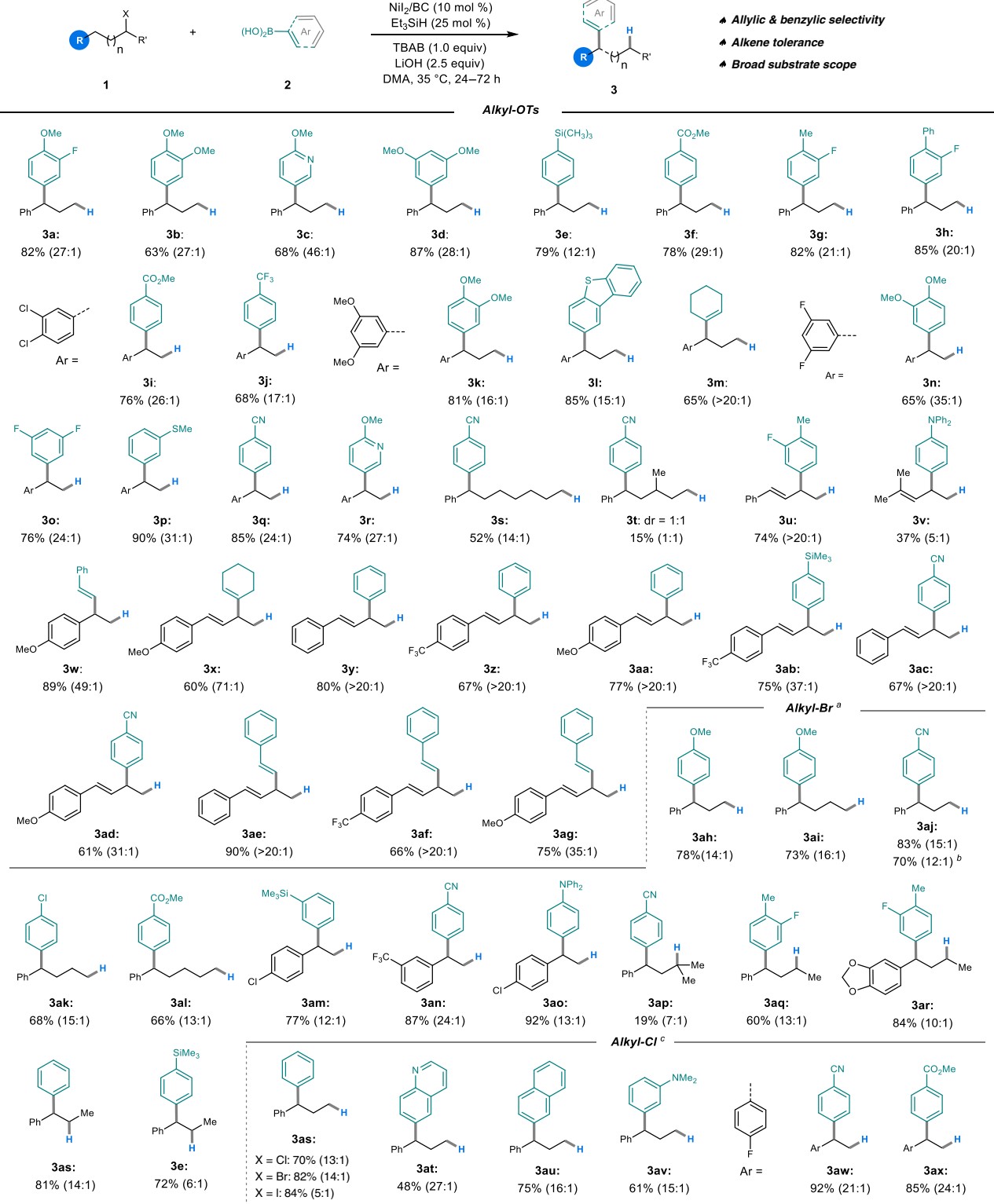

**Fig. 2 Scope of the methodology.** Isolated yields were shown. Ratios of **3:4** are determined by GCMS. [a]Without TBAB Additive. [b]10 mmol scale. [c]KI (0.38 mmol, 0.75 equiv), LiOH (0.75 mmol, 1.5 equiv), the mixture was stirred at 80 °C.

**Mechanism discussion.** The application of nickel catalyst constitutes a great advance in the cross-coupling of alkyl electrophiles[8,13]. The nitrogen-based ligands enable nickel-catalyzed reactions prefer to go through single-electron transfer (SET) processes, which increases the difficulty in mechanism investigation. Although preliminary mechanistic studies towards nickel-

catalyzed migratory cross-coupling have been done by Zhu's[24] and our group[26], several important concerns are still unclear, e.g., the origin of regioselectivity, the electronic character of nickel catalyst in chain-walking process and why olefins can dissociate from the catalyst during chain-walking, etc. This homogenous, redox-neural cross-coupling reaction provides a good platform to

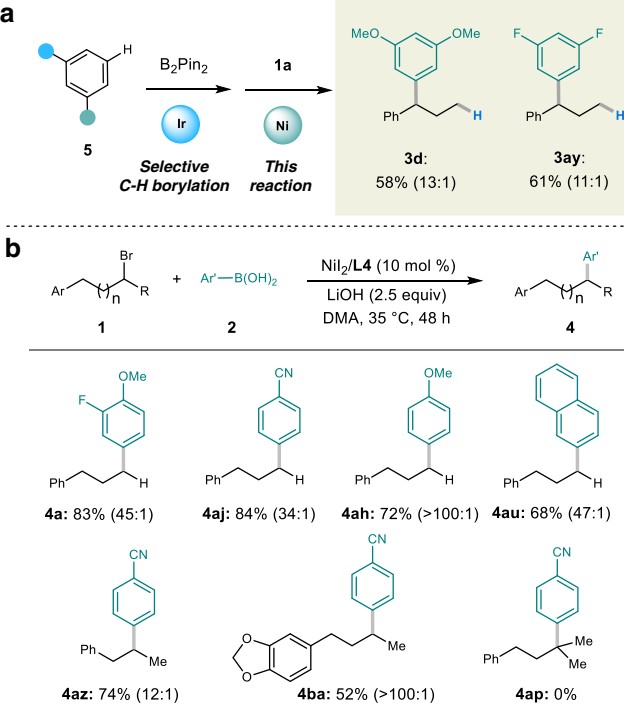

**Fig. 3 Expansion of substrate scope. a** Migratory Suzuki–Miyaura cross-coupling of aryl borate. **b** Original-site Suzuki–Miyaura cross-coupling.

obtain insights into the details of chain-walking, which would supply a rational basis for the further development of transformations involving nickel migration.

Moreover, this Suzuki–Miyaura reaction and our prior reductive condition reactions share many similarities, such as the same ligands, solvent, temperature, and the same selectivity (Fig. 4a). These similarities imply that these two type reactions share some commences in mechanism. The mechanistic study from the redox-neutral reaction can help us to understand the more complicated reductive ones.

Even though significant achievements have been made in mechanistic studies on nickel-catalyzed cross-coupling of alkyl electrophiles with an array of metal reagents[48,49] or another electrophiles[14]. However, reports on systematical mechanistic studies on the nickel-catalyzed Suzuki–Miyaura cross-coupling of alkyl electrophiles are still limited to date[50,51].

As a result, after achieving this nickel-catalyzed migratory Suzuki–Miyaura cross-coupling reaction, we started to investigate its mechanism.

First of all, to gain additional insight into the nickel migration, deuterium-labeled tosylates were synthesized and examined in this reaction. First, the reaction of **1a**-$D_2$ with **2b** furnished the D-migrated product **3bb**-$D_2$ with 100% benzylic deuterium retention and 63% D-incorporation at the homobenzylic position (Fig. 4b-1). These results suggest that the formation of migratory coupling products involve a key step of nickel chain-walking. Besides, the reaction of **1a**-$D_2'$ with **2c** delivered the migratory product **3as**-$D_2'$ with 98% deuterium retention under the standard conditions (Fig. 4b-2). These observations reflect that this method can serve as a platform to synthesize terminal partially D-labeled compounds. Deuterium-labeled molecules have been drawing increasing attention in medicinal chemistry recently[52,53], and partially D-labeled methyl groups ($CDH_2$ and $CD_2H$) were demonstrated to be useful in spectroscopic investigations and metabolic studies[54]. As a consequence, a series of 2D and 1D-labeled 1,1-diarylalkane derivatives were

prepared from the readily accessible α-D substituted tosylates (Fig. 4c).

Next, only trace amount of cross-coupling products were detected under the standard conditions when an alkyl bromide without an aryl group substituent (**6a**) was employed (Fig. 4d). This finding inspires us to investigate the role of the aryl group on the alkyl electrophiles. Further study found that using an electron-deficient olefin **L5** as ligand, the yield of the cross-coupling products could be improved to 30% (Fig. 4d). These results are agreed with the fact that reductive elimination from Ni (II) species is difficult, which can be promoted by the electron-deficient olefin ligands[32,55]. Therefore, the aryl group plays a significant role of promoting the reductive elimination.

Consequently, a catalytic cycle involving an alkyl-Ni(II)-Ar intermediate is proposed to rationalize this Suzuki–Miyaura reaction. As depicted in Fig. 4e, inefficient reductive elimination of the Ni(II) complex **III** leads to the formation of β-H elimination complex **IV**. Subsequent reductive elimination generates an arene (**8**) and an olefin (**9**).

The corresponding deborylation and olefin side products in the reaction of **6a** with **2c** were detected under our optimal conditions. A deuterium-labeled isopropyl bromide (**6b**-$D_6$) was examined under the standard conditions, the deborylation product (**8b**) was isolated in 47% yield with 31% D-incorporation (Fig. 4f)[56,57]. These results suggest that the hydrogen atom of the deborylation product is partly from the alkyl electrophile, which agrees with the proposed catalytic cycle (Fig. 4e) and reveals that the aryl group of the alkyl electrophile plays the role of promoting the carbon–carbon bond formation at benzylic position.

In addition, several reports demonstrated the possibility of reductive elimination from the Bn-Ni(II)-Ar intermediates to construct diarylalkanes[58], and a very recent report from the Fu group experimentally established the feasibility of Ni(II) chain-walking[59]. Based on these results, we proposed a catalytic cycle for the Ni-catalyzed migratory Suzuki–Miyaura reaction. As illustrated in Fig. 4g, the reaction is initiated by a Ni(0) species (**I**), which generates an alkyl-nickel(II) intermediate (**III**) by stepwise oxidative addition. There are two possibilities in the following steps: (A) a rapid chain-walking occurs to form a thermodynamically stable benzylic nickel(II) complex (**V**), and following transmetalation with a boronic acid to generate a benzyl-Ni(II)-Ar intermediate (**VI**); (B) transmetalation occurs before chain-walking, leading to the formation of the intermediate **VI** as well. Finally, the desired products **3** are produced from **VI** via reductive elimination.

These two pathways differ in the order of transmetalation and chain-walking. In order to differentiate these two pathways and to obtain details on the catalytic cycle, kinetic studies were conducted next.

**Kinetic studies**. To simplify the studies, we chose the reaction of alkyl bromide 1b with 4-cyanophenylboronic acid (2e) as a model reaction (Fig. 5a), which does not require the addition of silane. The dependence of the initial rate on nickel catalyst, alkyl bromide, boronic acid, and base concentration was evaluated in each case by monitoring the formation of product by gas chromatography. As illustrated in Fig. 5, a complex kinetic picture with first order in Ni at low catalyst loading and saturation kinetics at higher loadings was observed (Fig. 5b), possibly indicative of off-cycle pathways, which is also observed in a previous report.[60] The reaction exhibits a zeroth-order dependence on the concentration of alkyl bromide (Fig. 5c), and first-order dependence on the concentration of both aryl boronic acid (Fig. 5d) and base (Fig. 5e).

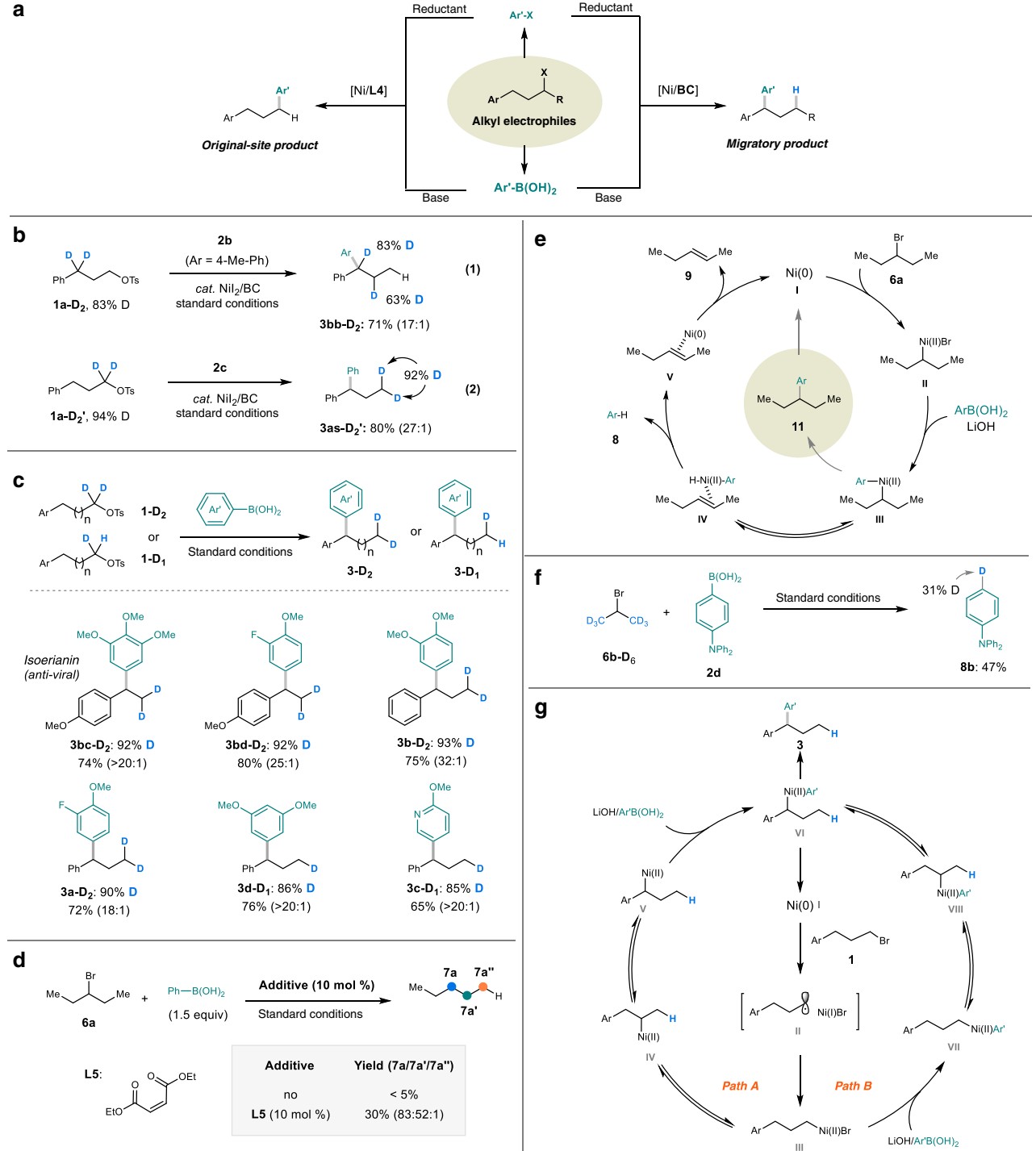

**Fig. 4 Mechanism discussion. a** Ni-catalyzed migratory cross-coupling of alkyl electrophiles with aryl coupling partners. **b** Experiments of D-labeled alkyl bromide with aryl group. **c** Synthesis of terminal D-labeled molecules. **d** Investigation the role of aryl group. **e** Proposed mechanism of D-labeled alkyl bromide without aryl group. **f** Experiments of D-labeled alkyl bromide without aryl group. **g** Proposed mechanism.

The above results reveal that: (1) the oxidative addition of alkyl bromide is a rapid and non-rate-determining step. A similar phenomenon was disclosed by Lei in the Ni-catalyzed oxidative coupling of arylzinc reagents[61]. (2) the transmetalation is the rate-determining step (RDS).

To further validate reaction pathway, relative rate constants were determined for the reaction of **2e** with different (p-RC₆H₄)(CH₂)₃Br (R=MeO, H, Cl, and CF₃). The reactions with different alkyl bromides were conducted independently. To our delight, plotting the $\log(k_R/k_H)$ values against the Hammett $\sigma_p$ constants resulted in a satisfactory linear correlation (Fig. 5f), and the electronic properties of the aryl groups on alkyl electrophiles indeed affect the reactivity. A positive $\rho$ value ($\rho = +0.45$) was detected, indicating that more electron-deficient benzyl-Ni(II) species show faster reactivity. This finding suggest the transmetalation occurs with a benzyl-Ni intermediate, which agrees with

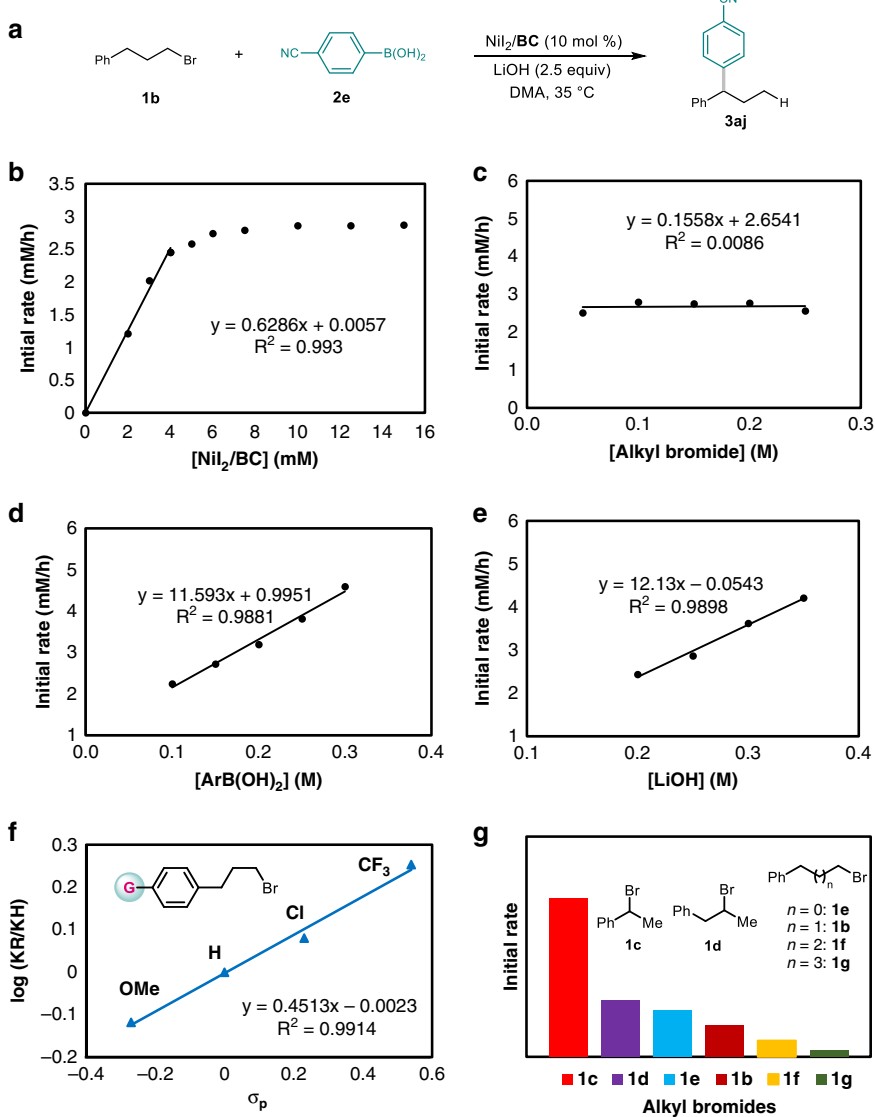

**Fig. 5 Kinetic profiles. a** Model reaction. **b** Reaction order in catalyst concentration. **c** Reaction order in alkyl bromide concentration. **d** Reaction order in aryl boronic acid concentration. **e** Reaction order in base concentration. **f** Hammett plots of alkyl bromides. **g** Initiate reaction rate of different alkyl bromides.

that the nickel chain-walking occurs prior to transmetalation, which is consistent with the path A, and rules out the path B (Fig. 4g).

Finally, the initiate rates of different alkyl bromides were also investigated (Fig. 5g). Among of these, secondary benzyl bromide (**1c**) exhibits the fastest rate, secondary unactivated alkyl bromide (**1d**) shows faster rate than the corresponding primary one (**1e**), which indicates the C–Br bond cleavage involving a SET process. Moreover, the alkyl bromide with longer carbon chain exhibits slower rate (compare **1e** with **1b**, **1f** and **1g**). These results indicate that chain-walking also affects the RDS, which is also agreed with path A and inconsistent with path B.

**Computational studies**. Extensive computational studies have been done on chain-walking process in the palladium-catalyzed reactions[62], in contrast, very limited reports on the nickel-catalyzed ones[63]. In order to get more details on the nickel chain-walking process of this reaction, density functional theory (DFT) calculations were carried out.

With the presence of **BC** ligand and reductant additive of $Et_3SiH$ species, the catalyst precursor $NiI_2$ would generate $Ni^0$ species by ligand exchange and reduction. We first calculated the oxidative addition of alkyl halides with Ni(0) species[64,65]. As shown in Fig. 6, active catalyst Ni(0) species **CP1** was chosen as the relative zero point for the free energy profiles, which is coordinated by a **BC** ligand and one solvent molecule ($N$, $N$-dimethylacetamide). Subsequent ligand exchange between the alkyl bromide and solvent molecule results in the formation of an intermediate **CP2**, with a large energy releasing (13.6 kcal/mol). The homolytic C–Br bond dissociation via **TS1** generates a 3-phenylpropyl radical **CP4** and an open-shell bromonickel(I) intermediate **CP3**. The radical rebounds to the nickel intermediate to produce a 3-phenylpropylnickel(II) intermediate **CP5**. The oxidative addition via a three-membered ring transition state **TS2**, needs to overcome a barrier of 22.0 kcal/mol, 7.5 kcal/mol higher than that of open-shell Ni(0)/Ni(I) pathway. The above results indicate that the alkyl-Ni(II) complex formation via a SET pathway is favorable.

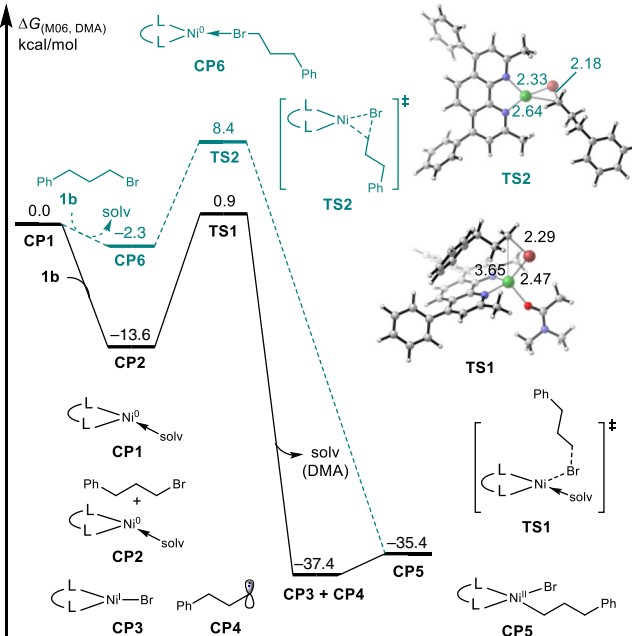

**Fig. 6 The calculated reaction energy profile of alkyl-Ni(II) complex formation steps.** Energy values are given in kcal/mol and represent the relative free energies calculated by the M06 method in *N,N*-dimethylacetamide. The values of bond length are given in ångstrom.

After the formation of 3-phenylpropylnickel(II) complex, the following chain-walking process by iterative β-hydride elimination and migratory insertion were taken into account. As illustrated in Fig. 7, the first β-hydride elimination process via **TS3** generates a planar Ni(II)-H species **CP7** and allylbenzene (**CP8**), which can insert to Ni–H bond to form a secondary C–Ni(II) species via **TS4**. Notably, the relative free energy of allylbenzene coordinated Ni(II) species **CP13** is 6.3 kcal/mol higher than **CP7**, which suggests that the olefin could dissociate from the nickel complex during the chain-walking process. Subsequent iterative β-hydride elimination and migratory insertion through **TS5** and **TS6** finally generates a 1-phenylpropylnickel(II) intermediate **CP12**. The calculated energy profile shows that the conversion of **CP5** to **CP12** is a reversible process: the highest energy barrier is first β-hydride elimination (14.5 kcal/mol) in the forward step (from **CP5** to **CP12**), and last alkene insertion (16.0 kcal/mol) in the reverse step (from **CP12** to **CP5**), with the same transition state **TS3**. Both the above two barriers are lower than that of transmetalation step (Fig. 8a, from **CP21** to **CP25** via the transition state **TS13**, with a barrier of 22.0 kcal/mol).

In addition, the cationic tetracoordinated nickel chain-walking process was also considered. In the cationic process, the initial state and final state (from **CP5** to **CP12**) are same as the ones in charge neutral process. The highest energy barrier changed to be the last alkene insertion (24.5 kcal/mol) in the forward process (from **CP5** to **CP12**) and the first β-hydride elimination (26.0 kcal/mol) in the reverse process (from **CP12** to **CP5**), with the same transition state **TS10**, which are higher than that in the neutral chain-walking process. Therefore, the cationic nickel chain-walking process is determined to be unfavorable. Energy profile of subsequent transmetalation and C–C bond formation is shown in Fig. 8b. Considering charge neutralization and coordination number, we choose the lithium phenylboronate **CP20**, which is generated via the coordination of hydroxyl to PhB(OH)₂, as active boron reagents (See Supplementary Fig. 55 for

details)[65,66]. Coordination between 1-phenylpropylnickel(II) intermediate **CP12** and lithium phenylboronate compounds **CP20** leads to the formation of **CP27**, which is able to generate a neutral Ni(II) complex **CP29** via a stepwise process involving transition states **TS16** (O–Li cleavage and coordinate Br–Li bond formation) and **TS17** (O–Ni formation and Ni–Br cleavage). Subsequent intramolecular transmetalation through a four-membered ring transition state **TS18** generates a phenyl-Ni(II)-benzyl intermediate **CP30**, which is exothermic by 27.0 kcal/mol. The energy barrier of transmetalation after chain-walking is 19.3 kcal/mol, which is 3.0 kcal/mol lower than that of direct transmetalation via transition state **TS13**. Finally, reductive elimination releases migratory product **3as** from **CP30**, via transition state **TS19** and with an energy barrier of 14.2 kcal/mol. The RDS is intramolecular transmetalation and the overall activation energy is 19.3 kcal/mol.

First of all, we found that the formation of the original-site coupling product was not concurrent with the migratory product (Fig. 9a, b), in other words, the regioselectivity varied during the reaction conduction (Fig. 9c). This finding reveals that the formation of the two products is probably from different catalytic cycles. Next, a radical probe experiment was carried out. The reaction of 6-bromo-1-hexene (**10**) with **2e** was studied in our reaction conditions. We found that the ratio of uncyclized cross-coupling product (**11U**) with the 5-exo-cyclized coupling product (**11R**) gradually increased along with the catalyst loading (Fig. 9d, e). These results indicate that a radical chain process is operating in this transformation, which is in accordance with the reductive cross-electrophiles coupling reactions[34].

Collectively, a catalytic cycle involving a radical chain process is proposed for the original-site Suzuki–Miyaura cross-coupling reaction. As depicted in Fig. 9f, the reaction is initiated by a Ni(II) species (**I**), and an aryl-Ni(II) species (**II**) is generated after transmetalation. The complex **II** reacts with an alkyl radical to generate a Ni(III) intermediate (**III**), which delivers the product (**4**) and a Ni(I) species (**IV**) via reductive elimination. The Ni(I) species **IV** reacts with an alkyl electrophile (**1**) to yield the alkyl radical and regenerate the Ni(II) catalyst.

Different ligands enable different regioselectivity, a phenomenon which also observed in our previous reductive conditions[26,27]. The mechanism of reductive cross-electrophile coupling of alkyl halides with aryl halides has been thoroughly studied by Weix and coworkers[34]. They have demonstrated that Ar–Ni(II) complex was able to react with the alkyl halide, initiating a radical chain process. Thereby, we think the reaction of Ni(II) complex [Ni]-**1** with the alkyl bromide (**1b**) is also the initiation of the radical chain process in our reaction (Fig. 9g). Moreover, in our prior reductive conditions, the electron-deficient aryl bromides could not efficiently yield the cross-coupling products, but homocoupling products (**12**) of the aryl bromides were observed instead. We conjecture that Ar–Ni(II) complexes [Ni]-**2** could also generate in the reactions with **BC** ligand, but [Ni]-**2** with **1b** was not able to trigger the radical chain, thus leading to the formation of **12** (Fig. 9h). In other words, the selectivity of the products derives from the reactivity of the nickel complexes towards the alkyl electrophiles. The less steric 5,5′-dmbpy (**L4**) is favorable for subsequent SET. Therefore, **L4** coordinated [Ni]-**1** is easier to generate the active Ni(III) compared with **BC** coordinated [Ni]-**2**. Instead, the Ni(0)/Ni(II) catalytic cycle operates in the reaction with **BC** as the ligand. If a stronger oxidant is used, it is supposed to overcome such steric effects and trigger the SET process. This is probably the reason that when more reactive alkyl iodides are used, poor regioselectivity was detected (Fig. 2, **3as**). Thus, the selectivity of the final products is determined by the ligand and the reactivity of the substrates.

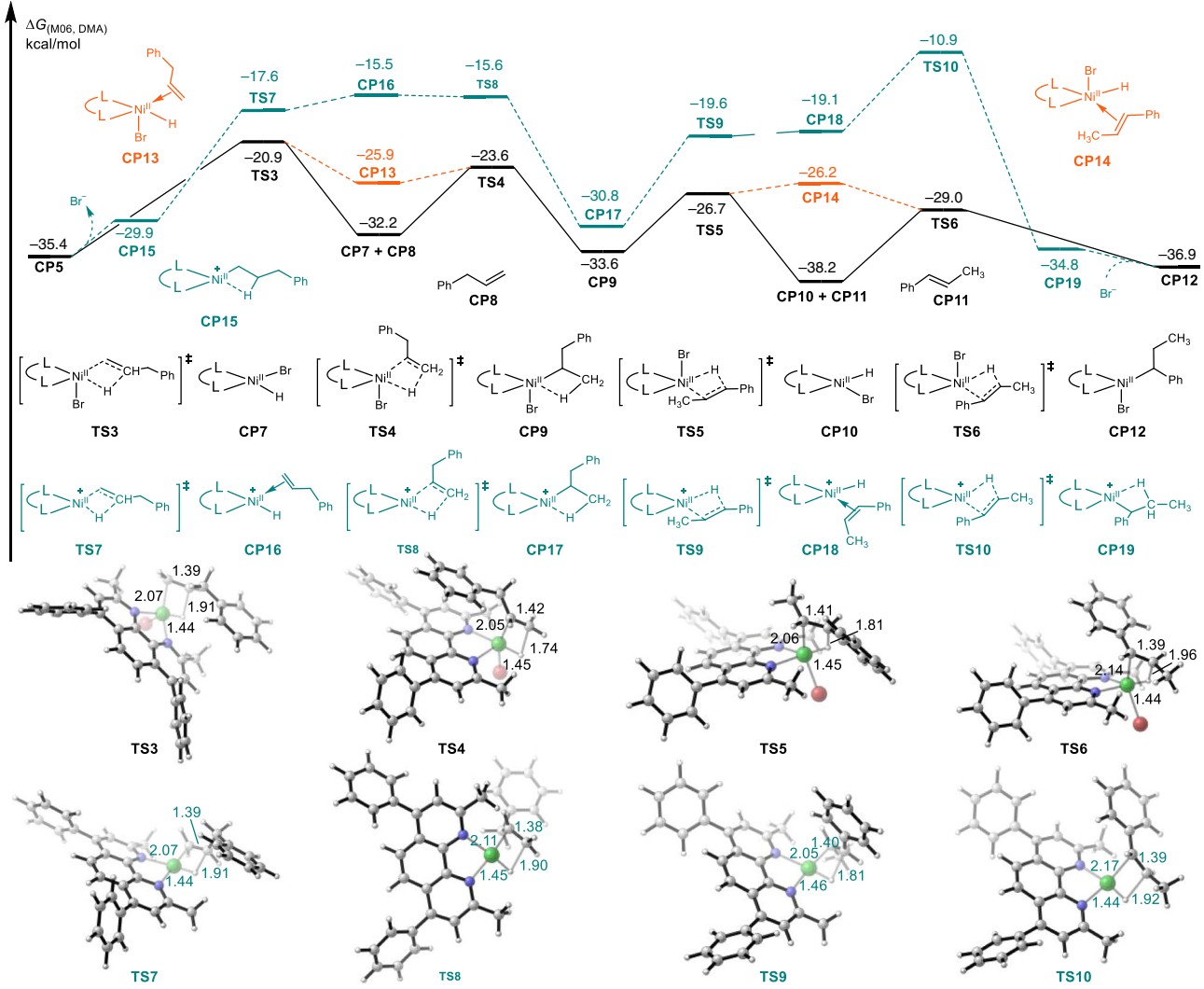

**Fig. 7 The calculated reaction energy profile of nickel chain-walking process.** Energy values are given in kcal/mol and represent the relative free energies calculated by the M06 method in N, N-dimethylacetamide.

## Discussion

In summary, we have developed a nickel-catalyzed migratory Suzuki–Miyaura cross-coupling reaction, featuring high benzylic or allylic selectivity. This method offers a protocol to rapidly access the diarylalkanes and allylbenzenes from a series of aryl/vinyl-bearing alkyl electrophiles and sp²-C boron-based nucleophiles. This approach is characterized by a broad substrate scope and excellent regioselectivity. In addition, the more challenging, but especially useful unactivated alkyl chlorides have been successfully used in the migratory cross-coupling. More potential applications in synthesis are addressed by the combination with selective borylation of C–H bonds. Furthermore, this protocol can serve as a platform to access terminal partially deuterium-labeled diarylmethine pharmacophores from the readily accessible precursors.

Mechanistically, the results of Hammett plots suggest that the transmetalation of boronic acid is with a benzyl-Ni(II) species. The DFT calculation suggest that oxidative addition of alkyl halides with Ni(0) species is prone to involve an SET pathway, the neutral nickel complex is more readily chain-walking than the cationic one, and the olefin is readily dissociated from the catalyst during chain-walking. In addition, the studies towards the original-site cross-coupling reveal that the formation of the product is likely to involve a radical chain process. Both Ni(0)/Ni(II) catalytic cycle and radical process are present in these reactions,

but the selectivity of the products is controlled by the catalyst and the substrate. We believe this study will greatly advance the future studies toward transformations involving nickel migration.

## Methods

**General procedure**. Under N₂ atmosphere, into an oven-dried 10 mL reaction tube equipped with a magnetic stir bar and sealed with a rubber stopper sequentially added NiI₂ (15.6 mg, 0.05 mmol, 10 mol%), **BC** (18.0 mg, 0.05 mmol, 10 mol%), anhydrous DMA (4 mL), and Et₃SiH (20 μL, 0.13 mmol, 25 mol%). The mixture was stirred at 35 °C for 30 min, then TBAB (161.2 mg, 0.5 mmol, 1.0 equiv), LiOH (29.9 mg, 1.25 mmol, 2.5 equiv), alkyl tosylate (0.5 mmol, 1 equiv), and aryl boronic acid (0.75 mmol, 1.5 equiv) were added in this order. The resulting mixture was stirred at 35 °C (if aryl boronic pinacol ester was used, stirred at 60 °C; if Alkyl bromide was used, TBAB was not need) and monitored by GC until the alkyl tosylate disappeared. After the reaction was complete, the mixture was quenched by saturated brine and extracted with ethyl acetate (20 mL × 3). The combined organic layers were dried over Na₂SO₄ and concentrated under reduced pressure. The resulting crude product was separated on a silica gel column affording the cross-coupling product.

## Data availability

The authors declare that the data supporting the findings of this study are available within the article and its Supplementary Information file. The raw data underlying Supplementary Figs. 3 with 55 are provided as a Supplementary Figures file. The raw data underlying Supplementary Tables 1 with 2 are provided as a Supplementary Tables file. The raw data for Absolute Calculation Energies, Enthalpies, and Free Energies are provided as a Source Data file. Additional data are available from the corresponding authors upon request.

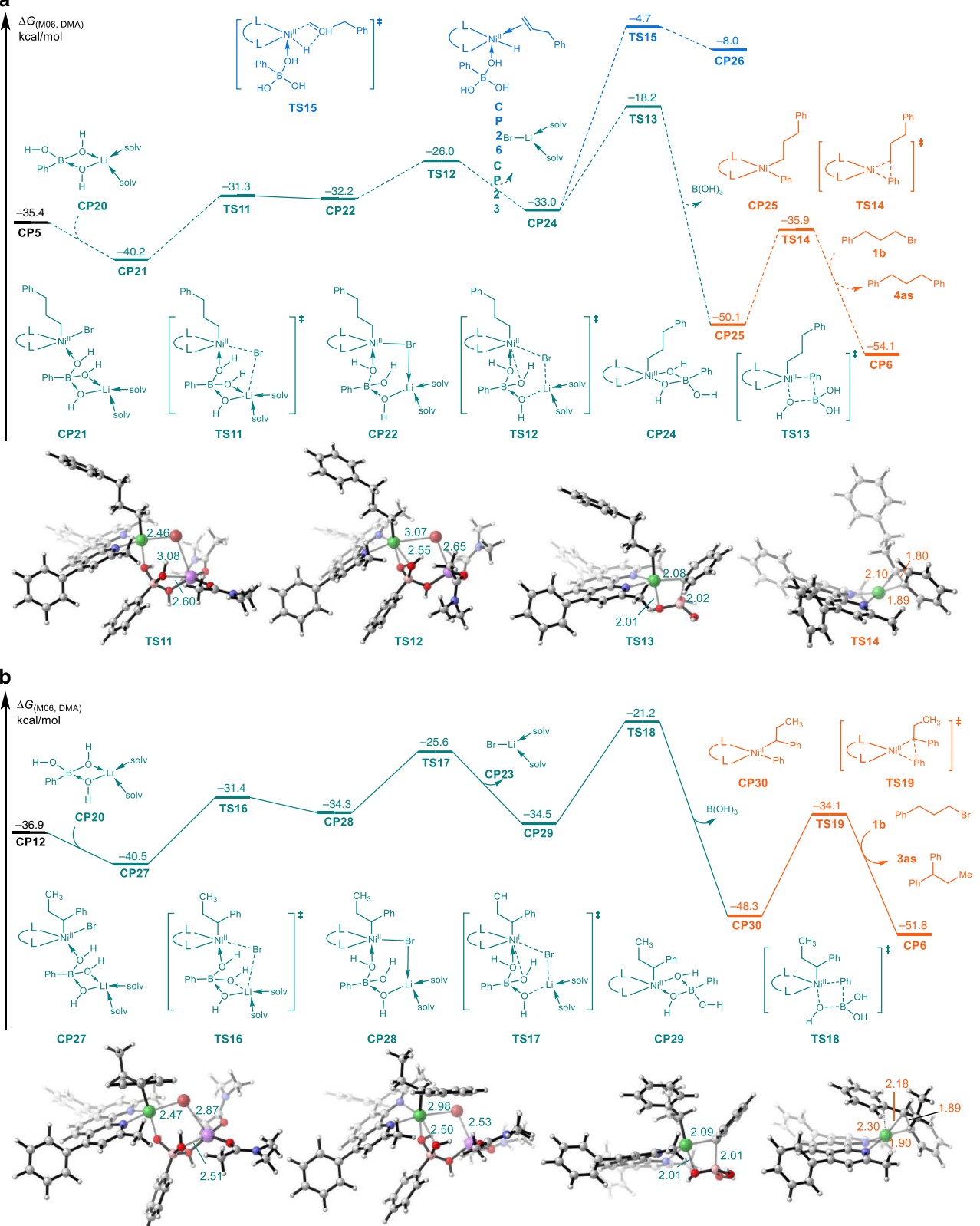

**Fig. 8 Calculated reaction energy profile. a** Chain-walking and original-site product formation process. Energy values are given in kcal/mol and represent the relative free energies calculated by the M06 method in $N$, $N$-dimethylacetamide. The values of bond length are given in ångstrom. **b** migratory product formation process. Energy values are given in kcal/mol and represent the relative free energies calculated by the M06 method in $N$, $N$-dimethylacetamide. The values of bond length are given in ångstrom.

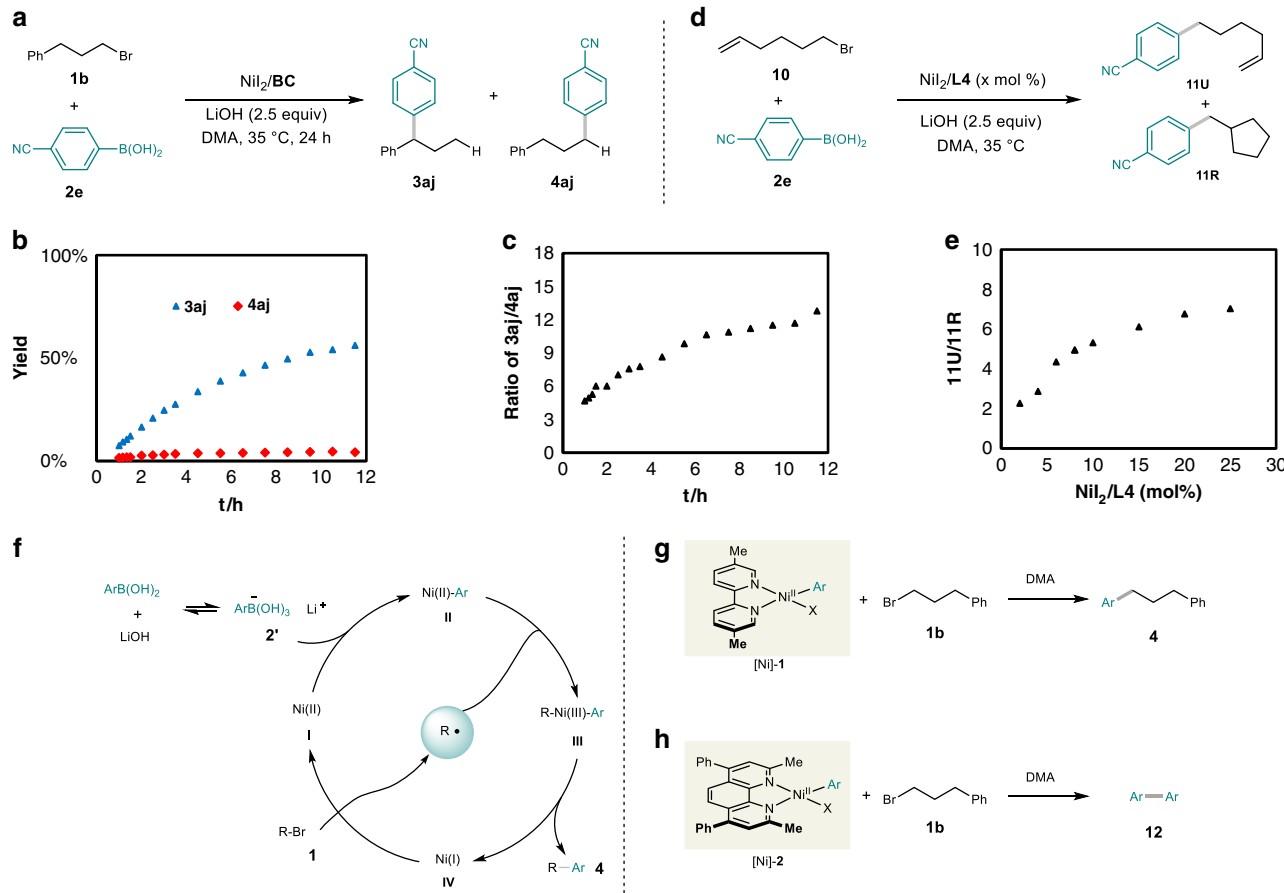

**Fig. 9 The study of original-site products. a** Model reaction. **b** Time course of product **3aj** and **4aj**. **c** Regioselectivity of migratory product with original-site product. **d** radical clock experiment. **f** Effect of catalyst loading on the ratio of **11U/11R**. **f** Proposed mechanism of original-site product. **g** Proposed process on reductive cross-coupling by Ni/**L4**. **h** Proposed process on reductive cross-coupling by Ni/**BC**.

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

## Acknowledgements

We thank Profs Qianghui Zhou, Wen-Bo Liu, and Aiwen Lei at Wuhan University for lending lab space and sharing the basic instruments. We are grateful for the financial support from National Natural Science Foundation of China (21702151, 21871211, 21822303, and 21703159) and the Fundamental Research Funds for Central Universities (2042019kf0208, 2018CDXZ0002 and 2018CDPTCG0001/4). We thank Dr Joshua Buss at University of Wisconsin-Madison for his very helpful suggestions on the manuscript preparation.

## Author contributions

Yuqiang Li designed and carried out most of the chemical reactions and analyzed the data. Long Peng, Yangyang Li, Binzhi Zhao, Wang Wang, Hailiang Pang, and Yi Deng supported the design and performance of synthetic experiments. Guoyin Yin designed and supervised the project. Yixin Luo, Ruopeng Bai, and Yu Lan conducted the calculations. Yuqiang Li, Yixin Luo, Ruopeng Luo, Yu Lan, and Guoyin Yin wrote the manuscript. All authors discussed the results and commented on the manuscript, and approved the final version of the manuscript.

## Competing interests

The authors declare no competing interests.
