## [Peer Review File · Nature Communications]

Reviewers' comments:

Reviewer #1 (Remarks to the Author):

Recommendation: Publish in nature communications after minor revisions.

Comments:

The full article by Yin and co-workers describes the Ni-catalyzed migratory Suzuki-Miyaura reaction with broad substrate scope and mechanistic studies. The process capitalizes on the ability of [Ni-H] catalyst to sustain migration over a hydrocarbon chain prior to a coupling at the remote benzylic or allylic position of the substrate.

The yields are usually good and the site selectivity is very high. The functional group tolerance is also very broad. The mechanistic investigations are well-conducted and offer novel and valuable insights on the emerging field of metal-catalyzed remote functionalization. Overall, this is an excellent contribution and once the modifications listed below have been taken into account it will undoubtedly meet the criteria of excellence set by the Nature Communications.

1. To give proper credit, the author should cite the Fu and Liu's hydroarylation work which was also catalyzed by nickel (Nat. Commun. 2016, 7, 11129);
2. In Figure 3, the descriptions about Figure 3D) and 3E) are missing.
3. To figure out if the NiH will dissociate from the olefin, the author should carry out extra experiments using two similar alkyl bromides while one of them is deuterated. It would support the dissociation if the migrated arylation products from the undeuterated alkylbromide is deuterated.
4. The authors should check the whole manuscript to make sure that all the compound numbers are bold. Please amend.

Reviewer #2 (Remarks to the Author):

In this manuscript, the authors describe a Ni-catalyzed Suzuki-Miyaura cross-coupling reaction between aliphatic electrophiles and aryl or alkenyl boronic acids. The ligand employed in the reaction has a large effect on the reaction outcome with one ligand causing the reaction to proceed with chain walking, while another ligand causes C-C bond formation to occur at the site of the original electrophile. The authors show that this reaction can occur with a broad range of substrates, providing access to diarylmethanes as well as aryl alkenyl methanes. The migration reaction itself can occur over long carbon chains (7 carbon atoms), accommodates a broad range of functional groups, and can employ primary, secondary and tertiary electrophiles. In addition to the impressively broad reaction scope, the authors have also conducted mechanistic experiments and computational experiments that provide very important information about the way this process operates.

Study toward transformation involving chain-walking is a very hot topic currently. The earth-abundant nickel has become a potential and powerful catalyst in this area. However, the mechanistic details are still quite limited. In this manuscript, the authors solved the regioselectivity by systematic and solid mechanistic studies. Very importantly, this work erased the gap between classical cross-coupling and migratory couplings. Another striking part of this work is the nickel catalyst exhibit better performance in chain-walking than palladium. Additionally, the results of theoretical calculations agree with the experiments and reveal that the chain-walking occurs at a five-coordinate intermediates rather than a cationic one and the olefins are readily to dissociate from the intermediates. All these results are significant contribution to the nickel chain-walking process. Overall, the manuscript is well-written, all experimental and computational studies have been well conducted to reveal the detailed reaction mechanism, and all references to important pioneering works have been quoted appropriately. I believe this work will be a highly impacted

paper, therefore I strongly support it to publish in Nature Communication with minor revision.

Comments:

1. The authors are smart to combine the D-Labelled experiments, kinetic studies, and DFT calculations from experiment and theory insights and detailedly explain the oxidative addition, transmetalation and chain-walking processes, then gives a reasonable catalytic mechanism of nickel-catalyzed migratory suzuki-miyaura cross-coupling reaction. Although this article is perfect in logic, before publishing on Nature Communications, it can be proofreaded again by English nativespeaker.
2. The deuterium labeling experiments in Figure 4B are quite interesting, I wonder to know the results of longer carbon chains.
3. Given DFT calculations reveal that alkene intermediates may dissociate from the catalyst, the authors should conduct an appropriate deuterium-labeling cross-over experiment that demonstrates the capacity for alkene dissociation.
4. The DFT calculations are a helpful addition to understanding the chain-walking and the regioselectivity. It's no doubt that the calculations of transmetallation process is well thought out by authours. For completeness' sake, I would recommend that they examine process of the Ni-Br and O-Li bond cleavage and Br-Li and Ni-O bond formation step in transmetallation is stepwise instead of collaborative.
5. Also, I recommend the authors add some other important references in computational parts of this work, eg. J. Am. Chem. Soc. 2014, 136, 1960; (about chain-walking), Angew. Chem. Int. Ed. 2019, 58, 3898; (the oxidized state change of nickel).
6. Using Hammett plots to study the boronic acid transmetalation process and finding the existence of benzyl-Ni(II) species is meaningful. But I think the Hammett plots of arylboronic acid should also be presented.
7. Choosing Ni(0) with two solvent molecules (N, N-dimethylacetamide) as 0.0 kcal/mol comparison and using lithium phenyl boronate compounds as boron reagents in transmetalation process in DFT calculations is feasible. But some description/explanation in these parts would be helpful.
8. If possible, giving three-dimensional images of the transition states in Figure 6 to better present the chain-walking process.

Response to Reviewer's Comments

Reviewer #1 (Remarks to the Author):

Recommendation: Publish in nature communications after minor revisions.

Comments:

The full article by Yin and co-workers describes the Ni-catalyzed migratory Suzuki-Miyaura reaction with broad substrate scope and mechanistic studies. The process capitalizes on the ability of [Ni-H] catalyst to sustain migration over a hydrocarbon chain prior to a coupling at the remote benzylic or allylic position of the substrate.

The yields are usually good and the site selectivity is very high. The functional group tolerance is also very broad. The mechanistic investigations are well-conducted and offer novel and valuable insights on the emerging field of metal-catalyzed remote functionalization. Overall, this is an excellent contribution and once the modifications listed below have been taken into account it will undoubtedly meet the criteria of excellence set by the Nature Communications.

We do thank this reviewer for his/her positive comments.

1) To give proper credit, the author should cite the Fu and Liu's hydroarylation work which was also catalyzed by nickel (Nat. Commun. 2016, 7, 11129).

Answer: We do thank the reviewer for this suggestion. The paper has been cited as ref 28 in the revised manuscript.

2) In Figure 3, the descriptions about Figure 3D) and 3E) are missing.

Answer: This error has been corrected in the revised manuscript. We do thank the reviewer to point out this error.

3) To figure out if the NiH will dissociate from the olefin, the author should carry out extra experiments using two similar alkyl bromides while one of them is deuterated. It would support the dissociation if the migrated arylation products from the undeuterated alkylbromide is deuterated.

Answer: According to this suggestion, we have carried out the experiment using two similar alkyl substrates while one of them is deuterated. First, the reaction of D-labelled alkyl tosylate with aryl boronic acid furnished the D-labelled product with 84% D-incorporation at the benzylic position and 38% D-incorporation at the benzylic position. Meanwhile, the reaction of undeuterated alkyl bromide with aryl boronic acid furnished the D-labelled product with 32% D-incorporation at the homobenzylic position. This result supports the dissociation of NiH from olefin. We have added the details in Supplementary Figure 26.

Supplementary Figure 26. D-labelled Experiment

We also have carried out extra experiments using olefins as additives. The results also support the dissociation of NiH from olefin. We have added the details in Supplementary Figure 22.

Supplementary Figure 22. Competition experiments of olefin additives

4) The authors should check the whole manuscript to make sure that all the compound numbers are bold. Please amend.

Answer: We thank the reviewer for this suggestion. All the compound numbers have been corrected.

Reviewer #2: (Remarks to the Author):

In this manuscript, the authors describe a Ni-catalyzed Suzuki-Miyaura cross-coupling reaction between aliphatic electrophiles and aryl or alkenyl boronic acids. The ligand employed in the reaction has a large effect on the reaction outcome with one ligand causing the reaction to proceed with chain walking, while another ligand causes C-C bond formation to occur at the site of the original electrophile. The authors show that this reaction can occur with a broad range of substrates, providing access to diarylmethanes as well as aryl alkenyl methanes. The migration reaction itself can occur

over long carbon chains (7 carbon atoms), accommodates a broad range of functional groups, and can employ primary, secondary and tertiary electrophiles. In addition to the impressively broad reaction scope, the authors have also conducted mechanistic experiments and computational experiments that provide very important information about the way this process operates.

Study toward transformation involving chain-walking is a very hot topic currently. The earth-abundant nickel has become a potential and powerful catalyst in this area. However, the mechanistic details are still quite limited. In this manuscript, the authors solved the regioselectivity by systematic and solid mechanistic studies. Very importantly, this work erased the gap between classical cross-coupling and migratory couplings. Another striking part of this work is the nickel catalyst exhibit better performance in chain-walking than palladium. Additionally, the results of theoretical calculations agree with the experiments and reveal that the chain-walking occurs at a five-coordinate intermediates rather than a cationic one and the olefins are readily to dissociate from the intermediates. All these results are significant contribution to the nickel chain-walking process. Overall, the manuscript is well-written, all experimental and computational studies have been well conducted to reveal the detailed reaction mechanism, and all references to important pioneering works have been quoted appropriately.

I believe this work will be a highly impacted paper, therefore I strongly support it to publish in Nature Communication with minor revision.

We do thank this reviewer for his/her positive comments.

Comments:

1) The authors are smart to combine the D-Labelled experiments, kinetic studies, and DFT calculations from experiment and theory insights and detailedly explain the oxidative addition, transmetalation and chain-walking processes, then gives a reasonable catalytic mechanism of nickel-catalyzed migratory suzuki-miyaura cross-coupling reaction. Although this article is perfect in logic, before publishing on Nature Communications, it can be proofreaded again by English native speaker.

Answer: We apologize for this problem. The language has been carefully corrected and modified by native English speaker.

2) The deuterium labeling experiments in Figure 4B are quite interesting, I wonder to know the results of longer carbon chains

Answer: According to this suggestion, we have prepared an alkyl tosylate with 4-C chain and tested the reaction. The result has been listed below and added in Supplementary Figure 25.

Supplementary Figure 25. D-labelled Experiment

3) Given DFT calculations reveal that alkene intermediates may dissociate from the catalyst, the authors should conduct an appropriate deuterium-labeling cross-over experiment that demonstrates the capacity for alkene dissociation.

Answer: According to this suggestion, we have carried out the experiment using two different alkyl substrates while one of them is deuterated. First, the reaction of D-labelled alkyl tosylate with aryl boronic acid furnished the D-labelled product with 84% D-incorporation at the benzylic position and 38% D-incorporation at the homobenzylic position. Meanwhile, the reaction of undeuterated alkyl bromide with aryl boronic acid furnished the D-labelled product with 32% D-incorporation at the homobenzylic position. The experiment supports the dissociation of NiH from olefin. We have added the details in Supplementary Figure 26.

Supplementary Figure 26. D-labelled Experiment

We also have carried out extra experiments using olefins as additives. The results also support the dissociation of NiH from olefin, and we have added the details in Supplementary Figure 22.

Supplementary Figure 22. Competition experiments of olefin additives

4) The DFT calculations are a helpful addition to understanding the chain-walking and the regioselectivity. It's no doubt that the calculations of transmetallation process is well thought out by authours. For completeness' sake, I would recommend that they examine process of the Ni-Br and O-Li bond cleavage and Br-Li and Ni-O bond formation step in transmetallation is stepwise instead of collaborative.

Answer: According to this comment, we examine the transmetallation process in Figure 8a and Figure 8b. We find that either in original-site or after migratory chain-walking, a ligand exchange step, including stepwise Li-Br bond generation and Ni-O bonds generation, could occur before transmetallation process. Coordination between alkyl-nickel intermediate **CP5** or 1-phenylpropylnickel(II) intermediate **CP12** and lithium phenyl boronate compounds **CP20** leads to the formation of five-coordinate nickel intermediates **CP21** or **CP27**. Subsequent ligand exchange process includes first O-Li bond cleavage and Br-Li bond formation (**TS11** or **TS16**), followed by second O-Ni formation and Ni-Br cleavage (**TS12** or **TS17**), to generate **CP24** and **CP29** respectively. Next, intramolecular transmetalation via **TS13** or **TS18** generates the phenyl-Ni(II)-alkyl intermediate **CP25** or phenyl-Ni(II)-benzyl intermediate **CP30**. We changed the corresponding data and description of the transmetallation process in our revised manuscript.

5) Also, I recommend the authors add some other important references in computational parts of this work, eg. J. Am. Chem. Soc. 2014, 136, 1960; (about chain-walking), Angew. Chem. Int. Ed. 2019, 58, 3898; (the oxidized state change of nickel).

Answer: We do thank the reviewer for this suggestion. These references have been added in the revised manuscript. Please see ref 62 for JACS 2014 and ref 64 for ACIE 2019.

6) Using Hammett plots to study the boronic acid transmetalation process and finding the existence of benzyl-Ni(II) species is meaningful. But I think the Hammett plots of arylboronic acid should also be presented.

Answer: We thank the reviewer for this suggestion. The Hammett plots of arylboronic acid have been carried out. The results show that the electronic properties of the aryl groups on arylboronic acid indeed affect the reactivity and more electronic-deficient aryl boronic acid shows faster reactivity, which also agrees with our proposed mechanism. And we have added the details in Supplementary Figure 41.

Supplementary Figure 41. Hammett plots of p-RC₆H₄B(OH)₂

7) Choosing Ni(0) with two solvent molecules (N, N-dimethylacetamide) as 0.0 kcal/mol comparison and using lithium phenyl boronate compounds as boron reagents in transmetalation process in DFT calculations is feasible. But some description/explanation in these parts would be helpful.

Answer: We thank the reviewer for this comment. Usually, the energy profile starts with the most stable catalyst species added/generated in the reaction system. In the experimental report, in the presence of BC ligand and reductive Et₃SiH species, the catalyst precursor NiI₂ would generate BCNi⁰ species by ligand exchange and reduction. As shown below, in the presence of solvent molecules (N, N-dimethylacetamide, DMA), solvent molecular coordination occurs to generate three or four-coordinated nickel species **CP1** or **CP1'**. The free energy of three-coordinated Ni⁰(BC)(DMA) **CP1** is 10.8 kcal/mol lower than four-coordinated Ni⁰(BC)(DMA)₂ **CP1'**. Therefore, **CP1** is considered to be the active catalyst species. We have changed the relative zero point in our revised manuscript and added this calculation in Supplementary Figure 55.

Supplementary Figure 55. Corresponding calculation of thermodynamic stability comparison between active catalyst CP1 and CP1'.

In our reaction, LiOH plays the role of activation of arylboronic acids. Considering charge neutralization and coordination number, we calculated the combined process of PhB(OH)₂ and LiOH·3DMA to generate CP20 and found 27.9 kcal/mol exoergicity, which indicates the possibility for the formation of lithium phenyl boronate compound CP20. Meanwhile, two relevant references (*J. Am. Chem. Soc.* **139**, 9909 (2017); *J. Org. Chem.* **82**, 9619 (2017)) are also cited as ref 65 and ref 66 at this part in our revised manuscript and this calculation has been added in Supplementary Figure 56.

Supplementary Figure 56. Calculation of the combine process of Ph-B(OH)₂ and LiOH·3DMA to generate CP20.

8) If possible, giving three-dimensional images of the transition states in Figure 6 to better present the chain-walking process.

Answer: Thanks for the reviewer's comment. We have added 8 geometrical structures (from TS3 to TS10) of the chain-walking process in Figure 7 in our revised manuscript.

REVIEWERS' COMMENTS:

Reviewer #1 (Remarks to the Author):

The revision is very thorough and detailed. The authors answered all of my questions and also those of other reviewers. I hope to be able to follow up on this chemistry in the future. The manuscript should be accepted in this form.

Reviewer #2 (Remarks to the Author):

I am satisfied for the revision and highly recommend its publication ASAP.